# Factors Associated with Polyp Detection Rate in European Colonoscopy Practice: Findings of The European Colonoscopy Quality Investigation (ECQI) Group

**DOI:** 10.3390/ijerph19063388

**Published:** 2022-03-13

**Authors:** Cristiano Spada, Anastasios Koulaouzidis, Cesare Hassan, Pedro Amaro, Anurag Agrawal, Lene Brink, Wolfgang Fischbach, Matthias Hünger, Rodrigo Jover, Urpo Kinnunen, Akiko Ono, Árpád Patai, Silvia Pecere, Lucio Petruzziello, Jürgen Ferdinand Riemann, Harry Staines, Ann L. Stringer, Ervin Toth, Giulio Antonelli, Lorenzo Fuccio

**Affiliations:** 1Digestive Endoscopy Unit and Gastroenterology, Fondazione Poliambulanza, 25124 Brescia, Italy; 2Digestive Endoscopy Unit, Università Cattolica del Sacro Cuore, 00168 Rome, Italy; silvia.pecere@gmail.com (S.P.); lucio.petruzziello@gmail.com (L.P.); 3Department of Medicine, OUH Svendborg Sygehus, 5700 Svendborg, Denmark; akoulaouzidis@hotmail.com; 4Department of Clinical Research, University of Southern Denmark (SDU), 5000 Odense, Denmark; 5Surgical Research Unit, OUH, 5000 Odense, Denmark; 6Department of Social Medicine and Public Health, Pomeranian Medical University, 70-204 Szczecin, Poland; 7Endoscopy Unit, IRCCS Humanitas Clinical and Research Center, 20089 Milan, Italy; cesareh@hotmail.com; 8Gastroenterology Department, Centro Hospitalar e Universitário de Coimbra, 3000-075 Coimbra, Portugal; pedro.amaro1967@gmail.com; 9Gastroenterology, Doncaster Royal Infirmary, Doncaster DN2 5LT, UK; anurag.agrawal1@nhs.net; 10Gastro Unit, Division of Endoscopy, Herlev and Gentofte Hospital, Copenhagen University, 2730 Herlev, Denmark; lene.brink@regionh.dk; 11Gastroenterologie und Innere Medizin, 63739 Aschaffenburg, Germany; wuk.fischbach@gmail.com; 12Independent Researcher for Internal Medicine, 97070 Würzburg, Germany; mhuenger@gmx.net; 13Instituto de Investigación Sanitaria ISABIAL—Servicio de Medicina Digestiva, Hospital General Universitario de Alicante, 03010 Alicante, Spain; rodrigojover@gmail.com; 14Department of Gastroenterology, Tampere University Hospital, 33521 Tampere, Finland; urpo.kinnunen@pshp.fi; 15Department of Gastroenterology, Hospital Clínico Universitario Virgen de la Arrixaca, 30120 Murcia, Spain; ono.akiko@gmail.com; 16Department of Gastroenterology and Medicine, Markusovszky University Teaching Hospital, 9700 Szombathely, Hungary; pataiarpaddr@gmail.com; 17Digestive Endoscopy Unit, Fondazione Policlinico Gemelli IRCCS, 00168 Rome, Italy; 18Department of Medicine C, Klinikum Ludwigshafen, 67063 Ludwigshafen, Germany; riemannj@garps.de; 19LebensBlicke Foundation, 67063 Ludwigshafen, Germany; 20Sigma Statistical Services Ltd., Saint Andrews KY16 0BD, UK; harry.j.staines@gmail.com; 21ECQI Secretariat, Buckinghamshire HP17 8ET, UK; ann.stringer@aspenmedicalmedia.com; 22Department of Gastroenterology, Skåne University Hospital, Lund University, 205 02 Malmö, Sweden; ervin.toth@med.lu.se; 23Department of Anatomical, Histological, Forensic Medicine and Orthopedics Sciences, Sapienza University of Rome, 00185 Rome, Italy; giulio.antonelli@gmail.com; 24Gastroenterology and Digestive Endoscopy Unit, Ospedale dei Castelli Hospital, 00040 Rome, Italy; 25Gastroenterology Unit, Department of Medical and Surgical Sciences, S.Orsola-Malpighi Hospital, 40138 Bologna, Italy; lorenzofuccio@gmail.com

**Keywords:** colonoscopy, colonoscopy standards, polyp detection rate, quality measures, survey

## Abstract

Background: The European Colonoscopy Quality Investigation (ECQI) Group aims to raise awareness for improvement in colonoscopy standards across Europe. We analysed data collected on a sample of procedures conducted across Europe to evaluate the achievement of the polyp detection rate (PDR) target. We also investigated factors associated with PDR, in the hope of establishing areas that could lead to a quality improvement. Methods: 6445 form completions from 12 countries between 2 June 2016 and 30 April 2018 were considered for this analysis. We performed an exploratory analysis looking at PDR according to European Society of Gastrointestinal Endoscopy (ESGE) definition. Stepwise multivariable logistic regression analysis was conducted to determine the most influential associated factors after adjusting for the other pre-specified variables. Results: In our sample there were 3365 screening and diagnostic procedures performed in those over 50 years. The PDR was 40.5%, which is comparable with the ESGE minimum standard of 40%. The variables found to be associated with PDR were in descending order: use of high-definition equipment, body mass index (BMI), patient gender, age group, and the reason for the procedure. Use of HD equipment was associated with a significant increase in the reporting of flat lesions (14.3% vs. 5.7%, *p* < 0.0001) and protruded lesions (34.7% vs. 25.4%, *p* < 0.0001). Conclusions: On average, the sample of European practice captured by the ECQI survey meets the minimum PDR standard set by the ESGE. Our findings support the ESGE recommendation for routine use of HD colonoscopy.

## 1. Introduction

The adenoma detection rate (ADR) is a validated quality measure that colonoscopists should constantly seek to improve. ADR requires the availability of histopathology results and can prove time-consuming to calculate [1]. The polyp detection rate (PDR) has been shown to be inversely associated with interval colorectal cancer (CRC) and, in the case of limited availability of histopathology data, can be used as a surrogate quality measure [2]. While there are concerns that the PDR can be manipulated, there is evidence to suggest that this may not be borne out in practice [1].

The European Colonoscopy Quality Investigation (ECQI) Group (www.ecqigroup.org accessed on 13 January 2022) comprises specialists and advisors and aims to raise awareness for improvement in colonoscopy standards across Europe. ECQI is a collaborative working party seeking cooperation and input from all involved in the field of colonoscopy. ECQI’s aim is not to create new quality criteria, but rather document dissemination of the European Society of Gastrointestinal Endoscopy (ESGE) guidelines and record their implementation in daily practice throughout Europe [3]. At the inaugural meeting of the ECQI Group, convened in 2013, to discuss quality in colonoscopy, the Group recommended devising a clinical practice questionnaire to evaluate the current practice of endoscopists across Europe. 

In 2017, the ESGE published performance measures for lower gastrointestinal endoscopy [4]. The ESGE recommend that endoscopy services across Europe adopt the key performance measures for measurement and evaluation in daily practice at a centre and endoscopist level. We analysed a sample of procedures conducted across Europe, between June 2016 and April 2018, in order to evaluate the current achievement of the PDR standard, as defined by the ESGE. We also analysed data collected on procedures with regard to factors associated with PDR, in the hope of establishing areas that could lead to an improvement in quality [4]. 

## 2. Materials and Methods

### 2.1. Questionnaire Development

The questionnaire was developed with consideration of the ESGE quality standards [5]. An iterative process was used to hone the questionnaire, ensuring that the time to complete the form was not too onerous [6]. This analysis uses the version finalized in 2016 and available for completion from 2 June 2016 (see Appendix A).

### 2.2. Recruitment

Participation was open to all Europe-based colonoscopists via web-based registration at the ECQI Group website. Awareness of the questionnaire came from abstracts, posters, presentations at national and international congresses, and individual communications from ECQI Group members. Interested participants applied via the ECQI Group website or to the ECQI Group Secretariat. Following verification, log-in access to the web-based questionnaire site was provided by email.

### 2.3. Ethics

This survey recorded routine practice and ethical approval was not generally required, but participants were advised to obtain ethical approval, if appropriate, according to their local protocols.

### 2.4. Dataset

Form completions from 2 June 2016 to 30 April 2018 were included in this analysis.

### 2.5. Polyp Detection Rate 

We performed an exploratory analysis looking at PDR according to ESGE definition [4]. All screening and diagnostic colonoscopies in patients aged 50 years or older were identified in our dataset using the reason for procedure question. ‘Signs and symptoms’ was classified as diagnosis and the responses, ‘Screening due to familial risk’, ‘Screening without pre-screening test’ and ‘Following positive screening test’ were classified as screening. The ‘Other’ responses free-text section was reviewed and responses re-classified as the above, as appropriate. All responses that were neither diagnosis nor screening were excluded from this analysis group. The presence of a polyp was regarded as positive if either ‘Polyp detected’ was ‘yes’ in any of the left, trans or right colon sections, or ‘Polypectomy (complete)’ or ‘Polypectomy (incomplete)’ were indicated under ‘Endoscopic intervention’.

### 2.6. Statistical Analysis

To preserve anonymity, only the patient’s year of birth was recorded. Age at the date of the procedure was derived assuming the date of birth was 30 June. Quantitative variables are presented as mean ± standard deviation (SD). Binary responses are presented as frequency and percentage. Where analysis was restricted by missing data, or in the case of subanalyses, the number of procedures included in the analysis is clearly stated.

Univariate binary logistic regression models were used to determine the association of individual variables with an endpoint using pre-defined categories. Stepwise multivariable logistic regression analysis was conducted to determine the most influential associated factors after adjusting for the other pre-specified variables. Stepwise analysis was performed on the following variables: age in 10-year categories; body mass index (BMI) categories; gender; inpatient status; reason for procedure; time of colonoscopy; previous total colonoscopy in last 5 years; sedation used; Boston Bowel Preparation Score (BBPS) ≥ 6; high-definition (HD) equipment used; assistive technology used; intended endpoint; and intended endpoint reached. Such analysis is restricted to the set of procedures for which all the pre-specified variables are known. 

No adjustment for multiplicity was made with a *p* value < 0.05 used to define significance. All analyses were performed using the statistical software package SAS version 9.4. (Cary, NC, USA).

## 3. Results

A total of 6445 completed procedure forms from 12 countries were considered for inclusion in analysis. Forms were received from 25 academic hospitals (*n* = 2270), 14 hospitals (*n* = 1235), 8 private institutions (*n* = 2657), 3 group practices (*n* = 160), and 1 other (*n* = 123). There were 3365 procedures that met the criteria for inclusion in the PDR analysis.

### Polyp Detection Rate

At least one polyp was detected in 40.5% (1363/3365) of qualifying procedures. The influence of the individual variables is shown in Table 1. The stepwise model, including 1448 procedures where the results of all selected variables were known, found use of HD equipment (*p* < 0.0001) to be the variable most associated with the PDR, followed, in order by BMI category (*p* < 0.0001), gender (*p* < 0.0001), age group (*p* < 0.0001) and the reason for the procedure (*p* = 0.0002) (Table 2). 

Within the screening category, PDR was highest in those receiving colonoscopy ‘Following positive screening test’ (60.4% of 460 procedures) compared with ‘Screening due to familial risk’ (43.4% of 235 procedures) and ‘Screening without pre-screening test’ (44.6% of 359 procedures) (*p* < 0.001).

Analysis of the number of polyps detected per patient, in the 1262 procedures with information on number of polyps recorded in all segments reporting ‘polyp detected’, found that 46.3% (584) had only one polyp reported (Figure 1). For these 1262 patients, the mean number of polyps per patient was 2.39 ± 2.53 with a median of 2.

For the 1363 procedures with a polyp detected, information on types of polyp detected was available for 1337 procedures. At least one flat lesion was reported in 373 procedures and at least one protruded lesion was reported in 1103 procedures. There were 2992 procedures that provided information on whether HD equipment was used. Analysis of the association of HD equipment with type of polyp detected shows that use of HD equipment significantly increases the proportion of procedures reporting flat lesions (14.3% vs. 5.7%, *p* < 0.0001). Use of HD equipment also significantly increased the proportion of procedures reporting protruded lesions (34.7% vs. 25.4%, *p* < 0.0001).

Analysis of use of HD equipment on polyp detection by segment showed a significant increase in all segments with the use of HD equipment: right 21.8% versus 13.7% (2902 procedures *p* < 0.0001); transverse 10.6% versus 6.7% (2909 procedures *p* = 0.0014); left 30.6% versus 18.6% (2959 procedures *p* < 0.0001) (Figure 2). 

In procedures where the total number of polyps were known by segment, there were 1009 polyps found in the right segment in 3223 procedures (data missing *n* = 142), 445 polyps in the transverse segment in 3242 procedures (data missing *n* = 123), and 1596 polyps in the left segment in 3263 procedures (data missing *n* = 102). Analysis of the total number of polyps detected according to use of HD equipment also showed a significant increase in the mean number detected using HD equipment in all segments: right 0.329 versus 0.193 (*p* < 0.0001); transverse 0.160 versus 0.073 (*p* = 0.0002); left 0.567 versus 0.311 (*p* < 0.0001). When considering only those procedures where at least one polyp is detected in a segment (i.e., excluding procedures with no polyp reported in that segment), the use of HD equipment significantly increased the mean number of polyps detected in the transverse segment only: right 1.574 versus 1.495 (*p* = 0.5965); transverse 1.558 versus 1.239 (*p* = 0.0121); left 1.938 versus 1.729 (*p* = 0.5660).

## 4. Discussion

In our sample, reflecting real-world practice, the PDR was 40.5%. This is comparable with the minimum standard suggested by the ESGE of 40% [4]. The variables found to be associated with PDR were in descending order: use of HD equipment, BMI, patient gender, age group, and the reason for the procedure.

Use of HD equipment was associated with a significantly higher PDR of 44.2% compared with 30.5% without the use of HD equipment (OR 1.802, *p* < 0.001). However, it cannot be concluded that HD equipment itself improves detection of polyps as there could be confounding factors, possibly related to clinician or clinic characteristics, associated with the use of HD equipment that have not been accounted for in our analysis. Nonetheless, our findings indicate that use of HD equipment is a more important factor than patient demographic variables such as age, gender, and BMI.

Review of the literature regarding the potential for HD equipment to improve polyp detection provides mixed results. Direct comparison of a HD endoscope with a standard definition (SD) endoscope in randomised trials has found a significant increase in polyp detection [7] or an increase in polyp detection that does not reach significance [8,9,10]. Meta-analysis of early studies found an increase in polyp detection with HD equipment (pooled incremental yield of HD over SD: 3.8%), but the clinical significance of this is not clear and there was no increase in the detection of high-risk adenomas [11]. However, recently, Roelandt et al. found an increase in detection of sessile serrated adenocarcinoma (8.2% vs. 3.2%) and adenocarcinomas (2.6% vs. 0.5%) with HD versus SD white light colonoscopy, despite no significant difference in overall ADR [12]. 

Retrospective studies have shown an increase in PDR in a ‘real-world’ setting [13,14,15,16], including after adjustment for confounding variables (age, sex, race, family history of colon cancer, history of polyps, aspirin use, indication for procedure, and endoscopist variables) [13]. Although another author showed no significant increase [17]. Interestingly, Waldman et al. analysed the difference in detection rates for endoscopists before and after the introduction of HD scopes [18]. They found that while overall there was a non-significant increase in both PDR and ADR, for those with a lower initial ADR (<20%) the introduction of HD scopes significantly increased their ADR (from 11.8% to 18.1%, *p* = 0.0026) [18]. 

Recent work by Pioche et al. and Zimmermann-Fraedrich et al. suggests that one reason for the apparent discrepancies in the evidence-base for HD equipment could be that at least a couple of generations of equipment advancement are required for differences in detection rates to become measurable [19,20]. However, these two studies conducted in different settings (hospital vs. private clinic) also produced contrasting results. It is, however, fair to conclude that detection rates tend to be higher with the use of more up-to-date equipment, as illustrated by a recent study in trainee endoscopists [21]. 

A recent network meta-analysis looking at a variety of colonoscopy modalities (SD, HD, narrow-band imaging, autofluorescence imaging, i-SCAN, Fuji Intelligent Color Enhancement—FICE, dye-based chromoendoscopy and novel image enhanced systems), found that only chromoendoscopy provided a significantly greater ADR than HD equipment [22]. For pair-wise comparison of PDR between SD and HD colonoscopy, while PDR was higher with HD equipment, it was not significant [22]. Another recent network analysis looking at additional improvements to HD colonoscopy found that the greatest improvement in ADR could be achieved by the addition of low-cost optimization techniques, such as water-aided colonoscopy, addition of a second observer, and dynamic position change [23]. This improvement would be greater than that provided by the addition of add-on devices, enhanced imaging techniques, or newer scopes, to HD colonoscopy.

We found that the use of HD equipment significantly increased the reporting of flat lesions (14.3% vs. 5.7%, *p* < 0.0001) and protruded lesions (34.7% vs. 25.4%, *p* < 0.0001). Di Caro et al. found a significant increase in flat polyp detection (18.2% vs. 5.2%, *p* < 0.001), but no significant change in protruded lesions (separated into pedunculated and sessile) with the use of HD equipment [24]. Rastogi et al. also found a significant increase in detection of flat adenomas with HD vs. SD equipment [25]. However, other investigations found no significant difference in detection of flat lesions [7,10,18], or indeed significantly higher flat polyp detection with SD rather than HD equipment [9]. 

Higher detection of polyps raised the potential of the ‘polyp paradox’ where patients undergo more frequent surveillance colonoscopies due to higher detection of polyps, which is based on more vigilant colonoscopists using better techniques, rather than increased incidence or risk level [26]. A longitudinal follow-up found that use of HD equipment versus SD at index colonoscopy did not result in any significant difference in subsequent interval adenoma or polyp detection, although there was a trend to lower detection rates in those who had a HD index colonoscopy [27]. We can report that whether the patient had had a total colonoscopy in the previous 5 years had no effect on the PDR in our dataset. However, we do not have information on the findings of the previous colonoscopy, who it was performed by, nor the techniques used.

In 2014, the European Society of Gastrointestinal Endoscopy (ESGE) recommended the “routine use of high definition white-light endoscopy systems for detecting colorectal neoplasia in average risk populations (weak recommendation, moderate quality evidence) [28].” This recommendation is augmented in the 2019 update, acknowledging the increase in the evidence-base, stating: “we can conclude that high definition systems may be of benefit to improve polyp and adenoma detection, although trial results are not entirely consistent [29]”.

Our findings support this recommendation, as the use of HD equipment was associated with improved polyp detection in our dataset, including a significant increase in the reporting of flat lesions.

### Limitations

The institutions and practitioners completing this questionnaire are from a range of countries across Europe. While this is a strength in that it provides a variety of practices enabling the assessment of a wide range of variables to see which influence outcomes, it also means that the practices in some countries may be over represented and could skew results. Selection bias is also possible as we have no control over which procedures practitioners choose to document. The term HD was not defined within the questionnaire, so we do not know exactly what equipment was used, and have relied on colonoscopist judgment as to whether they regarded their equipment as HD.

The publication of the ESGE performance measures occurred after this version of the questionnaire was compiled, therefore, there are some areas where our measures do not exactly match those specified by the ESGE. Some issues with the completion of the questionnaire were also identified, which have led to some responses being excluded from analysis due to either incorrect interpretation of the question or implausible responses. This has restricted the number of procedures that could be included in the multivariable analyses. In future, with a revised questionnaire design, we should be able to substantially reduce the number of non-eligible responses, which should improve the power of the analysis.

This analysis was exploratory in nature and guided by findings from these data. The observations would ideally be confirmed by prospective studies.

## 5. Conclusions

On average, the sample of European practice captured by the ECQI survey meets the minimum standard for PDR set by the ESGE. However, there is variation and potential for improvement. Our findings support the ESGE recommendation to routinely use HD white-light endoscopy systems. Of the variables assessed, the use of HD equipment was the most important factor associated with a higher PDR and was associated with a significant increase in the reporting of flat lesions.

## Figures and Tables

**Figure 1 ijerph-19-03388-f001:**
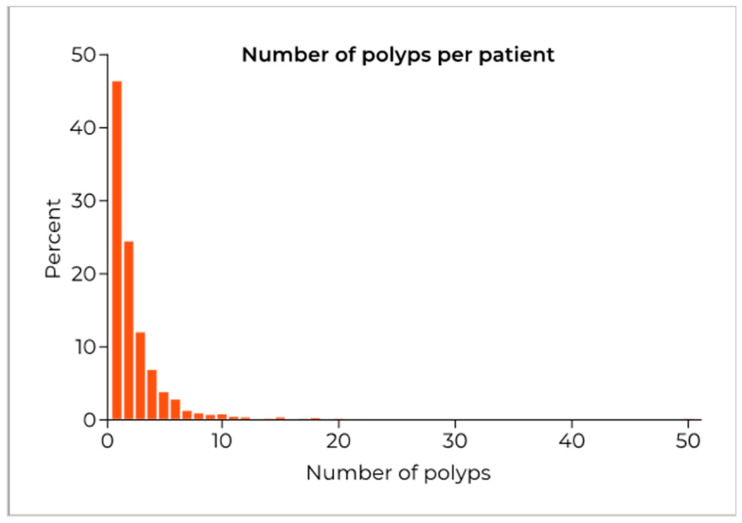
The distribution of the number of polyps per patient for the 1262 procedures that recorded the number of polyps in all segments with polyps detected.

**Figure 2 ijerph-19-03388-f002:**
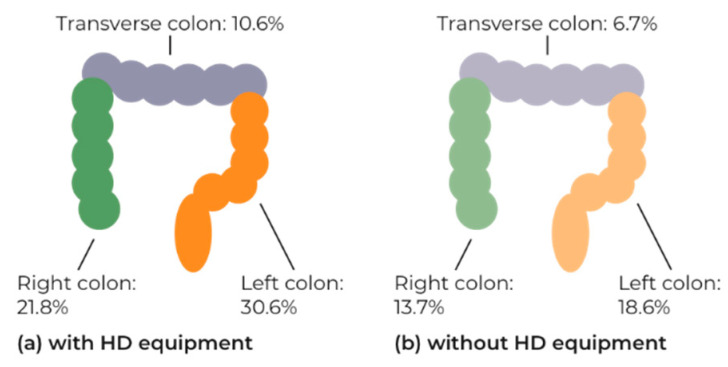
The proportion of procedures with a polyp detected by colon segment (**a**) with use of HD equipment and (**b**) without use of HD equipment. Includes procedures where both use of HD equipment and polyp detection by segment are known: right segment 2902 procedures; transverse 2909 procedures; left 2959 procedures.

**Table 1 ijerph-19-03388-t001:** Influence of individual variables on PDR.

Variable	Number with Polyp Detected (%)	Odds Ratio (95%CI)	*p* Value	*p* Value for Variable
**Age, years**				**<0.001**
50–59	319/1007 (31.7)	Reference		
60–69	556/1197 (46.4)	1.871 (1.571, 2.228)	<0.001	
70–79	385/905 (42.5)	1.597 (1.324, 1.925)	<0.001	
≥80	103/256 (40.2)	1.452 (1.094, 1.927)	0.01	
Missing data	None			
**BMI, kg/m^2^**				**<0.001**
<18.5	17/58 (29.3)	0.688 (0.386, 1.228)	0.206	
18.5–25	402/1069 (37.6)	Reference		
25–30	476/1040 (45.8)	1.400 (1.177, 1.666)	<0.001	
30–35	188/344 (54.7)	2.000 (1.564, 2.556)	<0.001	
>35	53/108 (49.1)	1.599 (1.075, 2.378)	0.021	
Missing data	746			
**Gender**				**<0.001**
Female	613/1771 (34.6)	Reference		
Male	750/1594 (47.1)	1.679 (1.461, 1.929)	<0.001	
Missing data	None			
**Patient type**				**0.368**
Outpatient	1043/2561 (40.7)	Reference		
Inpatient	166/432 (38.4)	0.908 (0.737, 1.120)	0.368	
Missing data	372			
**Reason for procedure**				**<0.001**
Diagnosis	821/2304 (35.6)	Reference		
Screening	542/1061 (51.1)	1.886 (1.628, 2.186)	<0.001	
Missing data	None			
**BBPS**				**0.246**
BBPS < 6 (inadequate)	172/465 (38.7)	Reference		
BBPS ≥ 6 (adequate)	1166/2805 (41.6)	1.129 (0.920, 1.386)	0.246	
Missing data	95			
**Previous total colonoscopy in last 5 years**				**0.914**
No	981/2431 (40.4)	Reference		
Yes	227/566 (40.1)	0.990 (0.821, 1.193)	0.914	
Missing data	368			
**Sedation used**				**0.019**
No	381/1012 (37.6)	Reference		
Yes	980/2334 (42.0)	1.199 (1.030, 1.395)	0.019	
Missing data	19			
**High-definition equipment used**				**<0.001**
No	250/820 (30.5)	Reference		
Yes	959/2172 (44.2)	1.802 (1.519, 2.139)	<0.001	
Missing data	373			
**Assistive technology used**				**0.015**
No	1100/2774 (39.7)	Reference		
Yes	136/289 (47.1)	1.353 (1.061, 1.725)	0.015	
Missing data	302			
**Colonoscopy endpoint**				**0.089**
Terminal ileum/neo terminal ileum	354/806 (43.9)	Reference		
Anastomosis	5/17 (29.4)	0.532 (0.186, 1.524)	0.24	
Caecum	997/2494 (40.0)	0.850 (0.724, 0.998)	0.048	
Missing data	48			
**Colonoscopy endpoint reached**				**<0.001**
No	37/159 (23.3)	Reference		
Yes	1309/3147 (41.6)	2.348 (1.615, 3.415)	<0.001	
Missing data	59			
**Time colonoscopy performed**				**0.622**
07:00–11:59 (morning)	504/1261 (40.0)	Reference		
12:00–17:59 (afternoon)	434/1035 (41.9)	1.085 (0.918, 1.282)	0.34	
18:00–19:59 (evening)	33/83 (39.8)	0.991 (0.630, 1.561)	0.97	
Missing data	986			

**Table 2 ijerph-19-03388-t002:** Results of stepwise analysis on 1448 procedures with all pre-specified variables known.

Rank	Variable	Response (*n*)	Proportion Reporting Polyp	Odds Ratio (95%CI)	*p* Value
1	Use of HD equipment	No (441)	30.8%	Reference	<0.0001
		Yes (1007)	52.7%	2.501 (1.973, 3.170)	
2	BMI category	<18.5 kg/m^2^ (33)	30.3%	0.632 (0.296, 1.350)	<0.0001
		18.5 < BMI ≤ 25 kg/m^2^ (618)	40.8%	Reference	
		25 < BMI ≤ 30 kg/m^2^ (565)	46.4%	1.256 (0.997, 1.581)	
		30 < BMI ≤ 35 kg/m^2^ (176)	63.6%	2.542 (1.797, 3.594)	
		>35 kg/m^2^ (56)	55.4%	1.801 (1.038, 3.124)	
3	Gender	Female (760)	40.7%	Reference	<0.0001
		Male (688)	52.0%	1.583 (1.286, 1.950)	
4	Age group	50–59 years (446)	36.1%	Reference	<0.0001
		60–69 years (540)	53.0%	1.993 (1.542, 2.576)	
		70–79 years (355)	48.2%	1.645 (1.238, 2.185)	
		≥80 years (107)	45.8%	1.495 (0.976, 2.291)	
5	Reason for procedure	Diagnosis (839)	40.9%	Reference	0.0002
		Screening (609)	53.2%	1.644 (1.322, 2.029)	

Stepwise analysis was performed on the following variables: age in 10-year categories; body mass index (BMI) categories; gender; inpatient status; reason for procedure; time of colonoscopy; previous total colonoscopy in last 5 years; sedation used; Boston Bowel Preparation Score (BBPS) ≥ 6; high-definition (HD) equipment used; assistive technology used; intended endpoint; and intended endpoint reached.

## Data Availability

An application form to access the data is available from the ECQI Secretariat upon receipt of a rationale and statistical analysis plan.

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
