# Peer review of "Factors Associated with Polyp Detection Rate in European Colonoscopy Practice: Findings of The European Colonoscopy Quality Investigation (ECQI) Group"

_ijerph, 2022, doi:10.3390/ijerph19063388_

Round 1
Reviewer 1 Report
This manuscript explores factors associated with polyp detection rate in European Colonoscopy Practice through stepwise multivariable logistic regression analysis. The aim of the manuscript was to evaluate the current practice of endoscopists across Europe. It was found that the use of HD equipment was associated with improved polyp detection. It may provide supports for the standard of colonoscopy in Europe. Unfortunately, this manuscript cannot be published in its current form. I recommend that authors consider the following points while editing the manuscript:
1). This manuscript analyzed 6445 completed procedures from 12 countries between 2 June 2016 and 30 April 2018. In addition, 3365 procedures met the criteria for inclusion in the PDR analysis. In table 1, the number for the BMI row is about 2600, there is a difference with 3365, and the reason is?.
2). P7, Lines 211-220. Nowadays, high-definition white-light endoscopy is widely used in clinical practice and can clearly observe the fine structure of tumors. However, the literature review of HD equipment in this section is not comprehensive.
3). The findings support the ESGE recommendation for routine use of HD colonoscopy. Because the ESGE recommendation was presented in 2014 and updated in 2019, new and prevalent colonoscopy modalities should be enrolled to evaluate. However, it cannot be completed in a short time. It is helpful to explain the reasons for using high-definition equipment and to highlight the novelty of the manuscript.
4). The lines format of Tables should be noted.
Author Response
Reviewer 1
|
This manuscript explores factors associated with polyp detection rate in European Colonoscopy Practice through stepwise multivariable logistic regression analysis. The aim of the manuscript was to evaluate the current practice of endoscopists across Europe. It was found that the use of HD equipment was associated with improved polyp detection. It may provide supports for the standard of colonoscopy in Europe. Unfortunately, this manuscript cannot be published in its current form. I recommend that authors consider the following points while editing the manuscript: |
|
|
REVEIWER COMMENTS |
RESPONSE |
|
1. This manuscript analyzed 6445 completed procedures from 12 countries between 2 June 2016 and 30 April 2018. In addition, 3365 procedures met the criteria for inclusion in the PDR analysis. In table 1, the number for the BMI row is about 2600, there is a difference with 3365, and the reason is?.
|
Missing data added to Table 1.
|
|
2. P7, Lines 211-220. Nowadays, high-definition white-light endoscopy is widely used in clinical practice and can clearly observe the fine structure of tumors. However, the literature review of HD equipment in this section is not comprehensive |
Any further comments from the Group?
We were looking specifically at papers comparing HD with SD for PDR.
If a significant paper is missing, please advise and it will be added.
It is not a comprehensive review of all literature regarding HD as that would be rather lengthy and not focused on how our findings contribute to existing knowledge. |
|
3. The findings support the ESGE recommendation for routine use of HD colonoscopy. Because the ESGE recommendation was presented in 2014 and updated in 2019, new and prevalent colonoscopy modalities should be enrolled to evaluate. However, it cannot be completed in a short time. It is helpful to explain the reasons for using high-definition equipment and to highlight the novelty of the manuscript. |
Data collected between 2016 and 2018 – did not collect information on why HD equipment used.
|
|
4. The lines format of Tables should be noted. |
Addressed |
Reviewer 2 Report
In my opinion, English language is understandable, and the work does not require any editing.
Authors noticed some interesting observations on the basis of data obtained from academic hospitals, hospitals, private institutions or group practices, that use of HD equipment is a more important factor for the polyp detection rate (PDR) than patient demographic variables such as age, gender and BMI.
Authors analyzed data collected on a sample of procedures (n=6445) conducted across Europe to evaluate the achievement of the polyp detection rate (PDR) target. Data of this manuscript is organized in the form of clear tables distinguishing the data into many categories. The PDR obtained in this research was 40.5%, comparable with the minimum standard suggested by the ESGE (40%).
The Authors describe that they obtained the results of 6445 completed procedure forms, but further describe the statistics based on 1363 cases from 3365 of qualifying procedures, later in Table 2 analysis of 1448 procedures, and the information below Tab.2 about the 1262 procedures with information on number of polyps reported. This can cause confusion for the Readers. My suggestion to Authors - without affecting the opinion and evaluation of the work- is to add information explain at the very beginning, that the statistics are based on surveys and cases classified for a specific statement, the study group appropriate to a given statistics.
The manuscript submitted by the Authors is in line with the subject of the International Journal of Environmental Research and Public Health, and will be an attractive article for the IJERPH Readers.
Author Response
Reviewer 2
|
In my opinion, English language is understandable, and the work does not require any editing. Authors noticed some interesting observations on the basis of data obtained from academic hospitals, hospitals, private institutions or group practices, that use of HD equipment is a more important factor for the polyp detection rate (PDR) than patient demographic variables such as age, gender and BMI. Authors analyzed data collected on a sample of procedures (n=6445) conducted across Europe to evaluate the achievement of the polyp detection rate (PDR) target. Data of this manuscript is organized in the form of clear tables distinguishing the data into many categories. The PDR obtained in this research was 40.5%, comparable with the minimum standard suggested by the ESGE (40%). |
|
|
REVEIWER COMMENTS |
RESPONSE |
|
1. The Authors describe that they obtained the results of 6445 completed procedure forms, but further describe the statistics based on 1363 cases from 3365 of qualifying procedures, later in Table 2 analysis of 1448 procedures, and the information below Tab.2 about the 1262 procedures with information on number of polyps reported. This can cause confusion for the Readers. My suggestion to Authors - without affecting the opinion and evaluation of the work- is to add information explain at the very beginning, that the statistics are based on surveys and cases classified for a specific statement, the study group appropriate to a given statistics. |
Statement added in 2.6 Statistical Analysis, as follows:
Where analysis was restricted by missing data, or in the case of subanalyses, the number of procedures included in the analysis is clearly stated.
Added to Table 2 legend with all pre-specified variables known Results of stepwise analysis on 1448 procedures with all pre-specified variables known
Added to Figure 2 legend Includes procedures where both use of HD equipment and polyp detection by segment are known: right segment 2902 procedures; transverse 2909 procedures; left 2959 procedures to Figure 2 title Figure 2. The proportion of procedures with a polyp detected by colon segment (a) with use of HD equipment and (b) without use of HD equipment. Includes procedures where both use of HD equipment and polyp detection by segment are known: right segment 2902 procedures; transverse 2909 procedures; left 2959 procedures. |
|
The manuscript submitted by the Authors is in line with the subject of the International Journal of Environmental Research and Public Health, and will be an attractive article for the IJERPH Readers. |
|
Reviewer 3 Report
General:
This article analysed data collected by the European Colonoscopy Quality Investigation (ECQI) group on a sample of procedures conducted across Europe to assess the current achievement of the PDR standard defined by the ESGE. They found that the following factors were associated with increasing PDR: use of high-definition equipment, body mass index (BMI), patient gender, age group, and reason for the procedure, in that order.
This is an interesting article that quantifies and points out the factors that increase the PDR.
Comments:
- Please explain in detail the definition of the term "high-definition", including the endoscope system, scope, and resolution that it represents.
- There are some papers that suggest that the PDR is related to the withdrawal time in a screening colonoscopy, however in this study, was it possible to compare the PDR with withdrawal time? Please explain.
- Was the use of HD equipment higher in academic hospitals? How many of the hospitals had both HD and SD equipment, and were there significant differences in PDRs within hospitals?
Were there any hospitals with SD equipment that had PDRs comparable to those of hospitals with HD equipment? Please present them.
- The authors described that a recent network meta-analysis looking at a variety of colonoscopy modalities (SD, HD, narrow-band imaging, autofluorescence imaging, i-SCAN, Fuji Intelligent Color Enhancement – FICE, dye-based chromoendoscopy and novel image enhanced systems), found that only chromoendoscopy provided a significantly greater ADR than HD equipment. I think some papers have reported that Linked Color Imaging (LCI) also increases PDR. Did the authors take into account chromoendoscopy and LCI use to PDR (ADR) in this study? Please explain.
Author Response
Reviewer 3
|
General: This article analysed data collected by the European Colonoscopy Quality Investigation (ECQI) group on a sample of procedures conducted across Europe to assess the current achievement of the PDR standard defined by the ESGE. They found that the following factors were associated with increasing PDR: use of high-definition equipment, body mass index (BMI), patient gender, age group, and reason for the procedure, in that order. This is an interesting article that quantifies and points out the factors that increase the PDR. |
|
|
REVEIWER COMMENTS |
RESPONSE |
|
Comments: 1.Please explain in detail the definition of the term "high-definition", including the endoscope system, scope, and resolution that it represents |
We did not ask for this information. The definition of HD is based on the practitioners assessment of what qualifies as HD. Added sentence to Limitations as follows:
The term HD was not defined within the questionnaire, so we do not know exactly what equipment was used, and have relied on colonoscopist judgment as to whether they regarded their equipment as HD. |
|
2. There are some papers that suggest that the PDR is related to the withdrawal time in a screening colonoscopy, however in this study, was it possible to compare the PDR with withdrawal time? Please explain.
|
ESGE definitions of PDR and withdrawal time differ in inclusion/exclusion criteria so it is not possible to directly relate them. A separate paper on withdrawal time has been published: Spada, C.; Koulaouzidis, A.; Hassan, C et al. Factors Associated with Withdrawal Time in European Colonoscopy Practice: Findings of the European Colonoscopy Quality Investigation (ECQI) Group. Diagnostics 2022, 12, 503. https://doi.org/10.3390/diagnostics12020503
While retraction time could have been included in our analysis it was felt that this would be prejudicial as in the event of finding a polyp and removing it the retraction time would inevitably increase. Analysis shows that mean retraction time with a polyp detected was 13.4 minutes and without a polyp detected it was 8.4 minutes. Retraction time was not defined, and therefore likely included time spent characterising and removing the polyp detected. Therefore, such findings are not informative as the role of cause versus effect cannot be established.
|
|
3. Was the use of HD equipment higher in academic hospitals? How many of the hospitals had both HD and SD equipment, and were there significant differences in PDRs within hospitals? Were there any hospitals with SD equipment that had PDRs comparable to those of hospitals with HD equipment? Please present them.
|
The proportion of procedures performed using HD equipment was highest in private practice (88%) followed by academic hospitals (82%), group practice (64%) and 23% in hospitals (the one other institution was 100% HD). Of those providing qualifying procedures (i.e. met PDR criteria and reported HD usage), 29 institutions used only one modality while 16 reported mixed usage. Of course, we do not know why there was mixed usage. It could be that HD equipment became available during the data collection period, colonoscopy discharge based on demand via different colonoscopes generations (common in several high-volume centres due to gradual capital equipment replacement) or even just due to discrepancies in form filling, for example. No institution registered sufficient number of procedures (over 100) to allow a reliable PDR estimate for both SD and HD usage. While we could provide PDR estimates for 2 institutions with SD usage and 6 institutions with HD usage, due to the number of confounding factors, we don’t believe that this would allow a meaningful comparison. In particular, we neither specify which procedures were to be documented nor we know if they were sequential; therefore, variations in procedures’ documentation could explain differences before you even get to patient profile or practitioner experience, etc. |
|
4.The authors described that a recent network meta-analysis looking at a variety of colonoscopy modalities (SD, HD, narrow-band imaging, autofluorescence imaging, i-SCAN, Fuji Intelligent Color Enhancement – FICE, dye-based chromoendoscopy and novel image enhanced systems), found that only chromoendoscopy provided a significantly greater ADR than HD equipment. I think some papers have reported that Linked Color Imaging (LCI) also increases PDR. Did the authors take into account chromoendoscopy and LCI use to PDR (ADR) in this study? Please explain.
|
ECQI Group please advise: Data collected between 2016 and 2018, was LCI available then (suspect not in widespread use)?
Data were collected between 2016 and 2018. At that time, LCI usage was not that prevalent and therefore was not included in our questionnaire. Data on chromoendoscopy use were collected; however, at the time of data collection, it was frequently used only upon the discovery of a polyp, or other reason for wishing to characterise more carefully, so it was not included in the analysis due to its likely prejudicial nature. In other words, we cannot provide accurate information if chromoendoscopy was used for polyp detection or for characterisation .
|
This manuscript is a resubmission of an earlier submission. The following is a list of the peer review reports and author responses from that submission.